# The Effects of Loving-Kindness Meditation Guided by Short Video Apps on Policemen’s Mindfulness, Public Service Motivation, Conflict Resolution Skills, and Communication Skills

**DOI:** 10.3390/bs15070909

**Published:** 2025-07-04

**Authors:** Chao Liu, Li-Jen Lin, Kang-Jie Zhang, Wen-Ko Chiou

**Affiliations:** 1School of Journalism and Communication, Hua Qiao University, Xiamen 361021, China; victory666666@126.com; 2General Education Center, Mindfulness Meditation Center, Ming Chi University of Technology, New Taipei 243303, Taiwan; lljen@mail.mcut.edu.tw; 3School of Education, City University of Macau, Macau, China; m24091100642@cityu.edu.mo; 4Department of Industrial Design, Chang Gung University, Taoyuan 33302, Taiwan

**Keywords:** loving-kindness meditation (LKM), mindfulness, public service motivation (PSM), conflict resolution skills (CRSs), communication skills (CSSs), police officers, short video app, digital intervention

## Abstract

Police officers work in high-stress environments that demand emotional resilience, interpersonal skills, and effective communication. Occupational stress can negatively impact their motivation, conflict resolution abilities, and professional effectiveness. Loving-Kindness Meditation (LKM), a mindfulness-based intervention focused on cultivating compassion and empathy, has shown promise in enhancing prosocial attitudes and emotional regulation. With the rise of short video platforms, digital interventions like video-guided LKM may offer accessible mental health support for law enforcement. This study examines the effects of short video app-guided LKM on police officers’ mindfulness, public service motivation (PSM), conflict resolution skills (CRSs), and communication skills (CSSs). It aims to determine whether LKM can enhance these psychological and professional competencies. A randomized controlled trial (RCT) was conducted with 110 active-duty police officers from a metropolitan police department in China, with 92 completing the study. Participants were randomly assigned to either the LKM group (n = 46) or the waitlist control group (n = 46). The intervention consisted of a 6-week short video app-guided LKM program with daily 10 min meditation sessions. Pre- and post-intervention assessments were conducted using several validated scales: the Mindfulness Attention Awareness Scale (MAAS), the Public Service Motivation Scale (PSM), the Conflict Resolution Styles Inventory (CRSI), and the Communication Competence Scale (CCS). A 2 (Group: LKM vs. Control) × 2 (Time: Pre vs. Post) mixed-design MANOVA was conducted to analyze the effects. Statistical analyses revealed significant group-by-time interaction effects for PSM (F(4,177) = 21.793, *p* < 0.001, η^2^ = 0.108), CRS (F(4,177) = 20.920, *p* < 0.001, η^2^ = 0.104), and CSS (F(4,177) = 49.095, *p* < 0.001, η^2^ = 0.214), indicating improvements in these areas for LKM participants. However, no significant improvement was observed for mindfulness (F(4,177) = 2.850, *p* = 0.930, η^2^ = 0.016). Short video app-guided LKM improves public service motivation, conflict resolution skills, and communication skills among police officers but does not significantly enhance mindfulness. These findings suggest that brief, digitally delivered compassion-focused programs can be seamlessly incorporated into routine in-service training to strengthen officers’ prosocial motivation, de-escalation competence, and public-facing communication, thereby fostering more constructive police–community interactions.

## 1. Introduction

The role of law enforcement officers, particularly policemen, is multifaceted and demanding ([58]; [96]). Policemen are often required to operate in high-stress environments where emotional resilience, mindfulness, and strong interpersonal skills are critical for effective performance ([22]; [74]). However, given the nature of their work, police officers are also at a heightened risk of experiencing occupational stress, burnout, and emotional exhaustion, which can impact their job satisfaction, motivation, and overall well-being ([73]). Therefore, enhancing mental resilience, emotional intelligence, and communication skills is essential not only for police officers’ personal development but also for sustaining effective public service, as these qualities are consistently linked to improved stress management, interpersonal functioning, and occupational performance in high-stress professions ([3]; [10]; [83]).

With the growing accessibility of digital technology, short video apps have emerged as powerful tools for delivering mental health interventions ([35]; [76]). These apps can guide mindfulness practices such as Loving-Kindness Meditation (LKM), a form of meditation rooted in generating compassion and empathy toward oneself and others ([51]; [59]). Research suggests that LKM can enhance mindfulness ([36]), emotional regulation ([53]), and prosocial behavior ([12]), making it a promising approach for law enforcement personnel who regularly engage with the public ([68]). Additionally, prior studies highlight that LKM promotes self-compassion, which is a key psychological mechanism that can mitigate occupational stress and enhance emotional resilience ([34]; [72]; [80]). Self-compassion has been shown to improve emotional regulation ([77]), reduce self-criticism ([43]) and foster greater interpersonal understanding ([79])—critical elements for police officers in managing public interactions under pressure.

Building on prior research demonstrating that Loving-Kindness Meditation (LKM) can enhance psychological functioning and prosocial behavior ([36]; [90]), this study seeks to examine the effectiveness of short video app-guided LKM interventions in improving key psychological and professional competencies among policemen, including mindfulness, public service motivation, conflict resolution skills, and communication skills—capacities known to be critical for effective policing under stress ([3]).

### 1.1. Loving-Kindness Meditation (LKM) and Its Psychological Benefits

Loving-Kindness Meditation (LKM) has been widely studied in the context of emotional regulation, empathy, and stress resilience ([51]). Research suggests that LKM fosters compassion and reduces negative affect, contributing to overall psychological well-being ([108]; [109]). Specifically, LKM fosters emotional regulation by helping individuals reframe negative affect and cultivate prosocial attitudes ([51]). In high-stress occupations—such as healthcare, where workers face life-or-death decisions and compassion fatigue, and law enforcement, where officers encounter acute threats and chronic organizational stress—mindfulness-based interventions, including LKM, have been associated with greater psychological resilience, lower physiological stress reactivity, and improved coping strategies ([3]; [22]).

Beyond its psychological benefits, LKM has also been found to influence interpersonal effectiveness and prosocial behavior. Studies indicate that engaging in LKM strengthens public service motivation and enhances individuals’ willingness to serve their communities ([42]; [89]). Furthermore, a growing body of research suggests that LKM enhances self-compassion, a psychological resource shown to buffer against burnout, increase job satisfaction, and strengthen interpersonal functioning in emotionally demanding professions such as policing and healthcare ([34]; [80]). These effects are particularly relevant in law enforcement, where officers are frequently required to regulate emotion under pressure and maintain high-quality public interactions. Self-compassion enables individuals to extend kindness toward themselves in stressful situations, reducing emotional exhaustion—a common challenge in law enforcement. By integrating self-compassion into LKM training, police officers may cultivate a more adaptive response to occupational stressors, leading to greater professional effectiveness.

Additionally, mindfulness-based practices have been associated with improved communication skills, particularly in high-pressure workplace environments ([89]; [93]). These findings suggest that LKM may serve as a valuable tool for improving emotional well-being and professional competencies in demanding occupational settings.

#### 1.1.1. Conceptual Definitions: Self-Compassion and Mindfulness

Self-compassion refers to an individual’s ability to treat oneself with kindness and understanding during times of failure or distress. It comprises three core components: self-kindness versus self-judgment, common humanity versus isolation, and mindfulness versus over-identification ([77]). Mindfulness, on the other hand, is defined as purposeful, present-moment awareness that is non-judgmental and accepting ([57]). While the two constructs are conceptually distinct, they are strongly interconnected—mindfulness facilitates awareness of suffering, and self-compassion provides an emotionally supportive response to that suffering.

Theoretically, self-compassion has been viewed as a construct that encompasses both mindfulness and a caring attitude toward oneself. Mindfulness provides the clarity of awareness, while self-compassion adds the dimension of warmth and care ([41]). Meta-analyses have found correlations between the two ranging from 0.55 to 0.70 ([70]), and evidence suggests that self-compassion mediates the beneficial effects of mindfulness on psychological well-being and occupational functioning ([26]). In high-stress occupations such as policing, mindfulness helps individuals recognize stressors, while self-compassion buffers against emotional exhaustion and fosters long-term motivation and prosocial behavior.

#### 1.1.2. Their Interrelationship and Empirical Evidence

Theoretically, self-compassion has been viewed as a construct that encompasses both mindfulness and a caring attitude toward oneself. Mindfulness provides the clarity of awareness, while self-compassion adds the dimension of warmth and care ([40]). Meta-analyses have found correlations between the two ranging from 0.55 to 0.70 ([70]), and evidence suggests that self-compassion mediates the beneficial effects of mindfulness on psychological well-being and occupational functioning ([24]). In high-stress occupations such as policing, mindfulness helps individuals recognize stressors, while self-compassion buffers against emotional exhaustion and fosters long-term motivation and prosocial behavior.

### 1.2. LKM in High-Stress Professions: Implications for Law Enforcement

While extensive research supports the psychological benefits of LKM, its application within law enforcement remains underexplored. Police officers frequently encounter high-pressure decision-making scenarios and interpersonal conflicts, making mindfulness-based interventions particularly relevant for this profession ([55]). Numerous studies in high-stress occupations such as healthcare and education have shown that mindfulness-based interventions, including LKM, can aid in emotion regulation, enhance decision-making, and improve workplace interactions ([15]; [54]). However, there remains a critical research gap regarding the applicability of LKM delivered via mobile or short video platforms specifically for law enforcement personnel. Given the unique stressors and public-facing responsibilities of police officers, this study addresses an urgent need to explore scalable, time-efficient interventions that can strengthen psychological resilience and professional competence in this population.

Despite these promising findings, little is known about the specific mechanisms through which LKM affects police officers’ public service motivation, communication skills, and conflict resolution abilities. Previous research has primarily focused on mindfulness interventions in general workplace settings, leaving a gap in understanding how such practices translate to law enforcement roles ([89]). Given the unique demands of policing—such as managing public interactions under stress, navigating ethical dilemmas, and resolving interpersonal conflicts—examining LKM’s impact within this context is critical for both theoretical development and practical applications ([42]; [85]).

### 1.3. Theoretical Framework and Mechanisms of LKM Impact

This study is grounded in Self-Determination Theory (SDT) and Social Learning Theory (SLT) to explain the mechanisms through which Loving-Kindness Meditation (LKM) may influence public service motivation (PSM), conflict resolution skills, and communication skills among police officers.

#### 1.3.1. Self-Determination Theory (SDT) and LKM

Self-Determination Theory posits that human motivation is driven by three fundamental psychological needs: autonomy, competence, and relatedness ([88]). In professional contexts, particularly in public service professions such as policing, intrinsic motivation plays a critical role in determining work engagement and interpersonal behaviors ([81]).

As a mindfulness-based intervention, LKM aligns with Self-Determination Theory (SDT) by promoting a sense of relatedness through compassion-oriented practices ([28]). Research has shown that LKM fosters kindness toward the self and others, reduces hostility and negative affect ([51]), and may enhance prosocial values such as public service motivation (PSM), which is critical for sustaining officers’ dedication to community service ([82]). Additionally, by promoting autonomy in emotional regulation and improving competence in managing interpersonal conflicts, LKM may indirectly improve officers’ conflict resolution skills and communication abilities ([86]).

Furthermore, SDT suggests that intrinsic motivation is more sustainable than extrinsic reinforcement in promoting behavioral changes ([13]). In this study, we hypothesize that LKM fosters long-term prosocial behavior by reinforcing internalized values of empathy and social responsibility, which are essential in law enforcement roles ([37]). Additionally, the integration of self-compassion into LKM aligns with SDT by fulfilling the need for emotional well-being, allowing officers to develop a stronger sense of self-efficacy and resilience when navigating professional challenges ([78]). This theoretical connection provides a strong justification for including public service motivation (PSM), conflict resolution skills, and communication skills as key outcome variables.

#### 1.3.2. Social Learning Theory (SLT) and LKM Delivery

Social Learning Theory emphasizes the role of observational learning, modeling, and reinforcement in shaping behavior ([9]). SLT is particularly relevant to the delivery method of LKM in this study, as the intervention was conducted through video-guided meditation sessions. According to SLT, individuals learn prosocial behaviors not only through direct experience but also by observing and imitating others who demonstrate such behaviors ([32]).

In the context of police training, structured mindfulness interventions such as LKM offer an opportunity for officers to observe, internalize, and replicate compassionate behaviors in their professional interactions ([46]). This aligns with previous research showing that mindfulness-based interventions delivered via digital platforms can effectively model prosocial behaviors and reinforce self-regulation strategies ([7]).

To maximize the effectiveness of the short video format used in our study, the LKM sessions were designed based on key principles of Social Learning Theory (SLT), incorporating guided modeling, reinforcement through repetition, and self-reflective practice ([99]). Guided modeling was implemented by providing explicit instructions on mindfulness techniques tailored for high-stress environments, allowing officers to observe and internalize effective coping strategies ([94]). Reinforcement through repetition was achieved by structuring the intervention into daily 10 min sessions and weekly 30 min deep practice sessions, ensuring consistent engagement and skill retention. Additionally, self-reflective practice was encouraged through post-meditation journaling, enabling participants to reinforce learned concepts and enhance metacognitive awareness ([92]). By integrating both Self-Determination Theory (SDT) and SLT, our study builds on existing psychological frameworks to explain how LKM fosters intrinsic motivation (SDT-based) and behavioral learning (SLT-based), ultimately contributing to improvements in public service motivation, conflict resolution skills, and communication abilities among police officers ([11]). In particular, this study extends existing research by incorporating self-compassion as a key psychological mechanism underlying the benefits of LKM, examining its potential role in buffering stress and enhancing officers’ professional competencies.

#### 1.3.3. Theoretical Framework

Based on the above theoretical rationale, we conceptualize self-compassion as a key psychological mechanism in the pathway from mindfulness to enhanced professional functioning. In this study, the LKM-based short video intervention is expected to first activate mindfulness and then enhance self-compassion, which in turn is hypothesized to reduce emotional exhaustion and improve prosocial motivation and communication efficacy. We also present the correlation matrix of self-compassion and outcome variables in Section 3 to support this model.

### 1.4. Purpose of the Study

The purpose of this study is to investigate the effects of LKM guided by short video apps on policemen’s mindfulness, public service motivation, conflict resolution skills, and communication skills. By examining these outcomes, this study aims to contribute to the growing literature on mindfulness interventions in high-stress professions and to explore scalable solutions for improving the well-being and interpersonal effectiveness of police officers. In particular, this study extends existing research by incorporating self-compassion as a key psychological mechanism underlying the benefits of LKM, examining its potential role in buffering stress and enhancing officers’ professional competencies.

Based on the literature and theoretical considerations, we propose the following hypotheses:

**Hypothesis 1 (H1).** *LKM intervention will significantly improve participants’ mindfulness*.

**Hypothesis 2 (H2).** *LKM intervention will significantly improve participants’ public service motivation*.

**Hypothesis 3 (H3).** *LKM intervention will significantly improve participants’ conflict resolution skills*.

**Hypothesis 4 (H4).** *LKM intervention will significantly improve participants’ communication skills*.

## 2. Method

### 2.1. Participants

The participants in this study were active-duty police officers from a metropolitan police department in China (Figure 1). Recruitment information was disseminated through the department’s internal communication channels, where the study was described as a stress-relief and self-awareness program aimed at enhancing compassion, mindfulness, and interpersonal skills ([7]). A total of 200 active-duty police officers from a metropolitan police department in China were invited to participate in this study. These officers were selected through convenience sampling, using the department’s internal communication channels. Of the 200 invited, 110 officers voluntarily agreed to join, and 92 completed both pre- and post-intervention assessments. The sample was composed of 71.7% males (n = 66) and 28.3% females (n = 26), with an average age of 35.22 years (SD = 3.19).

Inclusion criteria required participants to be active-duty officers aged 21 or older and fluent in Mandarin Chinese. Exclusion criteria included a history of diagnosed mental health disorders (such as major depression, anxiety disorders, or post-traumatic stress disorder (PTSD) and prior experience with meditation practices ([21]).

To better understand the demographic characteristics of the sample, we collected additional background information. Participants had varying years of police service: 30% had 1–5 years of experience, 45% had 6–10 years, and 25% had over 10 years of service. Regarding prior experience with mindfulness practices, 25% of officers reported having engaged in mindfulness activities such as breath meditation or mindful walking before the study, while 75% had no prior exposure to mindfulness training ([75]). Given the voluntary nature of participation, we acknowledge that self-selection bias may be present, potentially influencing the representativeness of the sample ([102]). Future research should aim to include a broader range of participants to enhance generalizability.

A total of 92 participants completed the intervention and were randomly assigned to either the LKM intervention group (n = 46) or the control group (n = 46). The demographic details of the participants are presented in Table 1. No significant differences were found in age, gender, or other demographic factors between the two groups ([106]).

### 2.2. Instruments

The questionnaire used in this study consisted of four main sections: mindfulness, public service motivation, conflict resolution skills, and communication skills. All variables were measured using established scales, with minor adaptations to fit the context of police work. Participants responded to each item using a 5-point Likert scale (1 = strongly disagree to 5 = strongly agree). To ensure the accuracy and validity of the questionnaire, a strict back-translation process was conducted, following established protocols to adapt the scales for Chinese-speaking participants.

#### 2.2.1. Mindfulness Scale

Mindfulness was assessed using the Mindfulness Attention Awareness Scale (MAAS), originally developed by [14] ([14]) and adapted for this study to reflect relevant aspects of mindfulness in police work ([14]). The scale consists of 15 items that measure participants’ open and receptive awareness of present-moment experiences. All items were rated on a 5-point Likert scale ranging from 1 (“Almost Never”) to 5 (“Almost Always”), with higher scores indicating a higher level of dispositional mindfulness. The MAAS demonstrated strong internal consistency in the current study (Cronbach’s α = 0.89).

The MAAS was selected for this study due to its unidimensional focus on present-moment attention and awareness, which aligns closely with the core cognitive processes most relevant to law enforcement contexts. Compared to multidimensional measures like the Five Facet Mindfulness Questionnaire ([6]), the MAAS is more concise and has demonstrated strong psychometric properties in occupational populations, including those experiencing high levels of stress. Additionally, its singular structure allows for cleaner integration into structural models, particularly those involving meditation.

#### 2.2.2. Public Service Motivation Scale

Public service motivation was measured using the Public Service Motivation Scale (PSM), initially developed by [25] ([25]) and modified to assess motivation specific to public service roles ([25]). The scale includes 10 items covering three core dimensions: compassion, self-sacrifice, and commitment to the public good. All items were rated on a 5-point Likert scale ranging from 1 (“Strongly Disagree”) to 5 (“Strongly Agree”), with higher scores indicating stronger public service motivation. The PSM demonstrated excellent internal consistency in this study (Cronbach’s α = 0.91).

#### 2.2.3. Conflict Resolution Skills Scale

Conflict resolution skills were measured using the Conflict Resolution Styles Inventory (CRSI). The scale was designed by Kurdek for the policing environment of this study ([62]). The scale includes 4 items that evaluate participants’ abilities to manage and de-escalate conflicts, with a focus on empathy, active listening, and collaborative problem-solving. All items were rated on a 5-point Likert scale ranging from 1 (“Strongly Disagree”) to 5 (“Strongly Agree”), with higher scores indicating stronger conflict resolution skills. The CRSI demonstrated satisfactory internal consistency in this study (Cronbach’s α = 0.87).

#### 2.2.4. Communication Skills Scale

Communication skills were assessed using the Communication Competence Scale (CCS). The scale was designed by [104] ([104]) and adapted for use with police officers to measure essential communication skills such as clarity, empathy, and effectiveness in interactions with the public ([104]). The scale consists of 36 items assessing dimensions such as clarity, empathy, responsiveness, and effectiveness in interactions with the public. All items were rated on a 5-point Likert scale ranging from 1 (“Strongly Disagree”) to 5 (“Strongly Agree”), with higher scores indicating greater communication competence. The CCS demonstrated excellent internal consistency in the current study (Cronbach’s α = 0.93).

### 2.3. Short Video-App-Guided Loving-Kindness Meditation Intervention

The intervention consisted of a series of short video sessions focused on Loving-Kindness Meditation (LKM), delivered over a 6-week period via a mobile app. Each session was structured to guide participants through practices designed to cultivate compassion, empathy, and emotional awareness, with a particular focus on enhancing interpersonal skills relevant to police work. The intervention was developed in collaboration with mindfulness experts and adapted specifically for the needs of law enforcement professionals ([51]).

In addition to the guided LKM sessions, the app also included supplementary content aimed at enhancing participants’ understanding and engagement with mindfulness principles. Specifically, the app provided psychoeducational modules that introduced the theoretical background and benefits of LKM in law enforcement contexts, emphasizing its role in emotional regulation, stress management, and interpersonal communication ([91]). Furthermore, participants were encouraged to engage in self-reflection exercises following each meditation session, which involved digital journaling prompts designed to reinforce key insights and track personal progress ([78]). Additionally, the app incorporated interactive elements such as automated reminders, motivational messages, and access to a facilitator for inquiries and guidance. These features aimed to create a more immersive and supportive learning experience, reinforcing the core objectives of the intervention beyond the direct meditation practice ([14]).

LKM follows a series of steps where the practitioner imagines the object of loving-kindness and extends four blessings: wishing them to have no enemies, wishing them to experience no pain, wishing them to have no illness, and wishing them to possess happiness. The typical sequence of target selection for LKM follows this order: first, bless yourself; then bless those you like; next, bless neutral individuals; followed by blessing those who make you angry; and finally, equally bless yourself, those you like, neutral people, and those you dislike, extending blessings to all people.

Participants in the control group were placed on a waitlist and continued their regular patrol and administrative duties throughout the four-week intervention period. They did not receive any mindfulness or emotional skills training. However, to minimize differences in researcher contact across conditions, the control group received weekly reminder emails that contained only neutral administrative content and no psychological materials. All control participants completed the same pre- and post-intervention assessments as the experimental group.

#### 2.3.1. Session Structure and Content

The short video application of LKM consisted of animation-guided LKM to conduct the online LKM intervention training. Participants were asked to do the following: (1) Psychoeducation on LKM (5 min): Sessions began with a brief introduction to the psychological foundations of LKM, emphasizing its benefits for emotional regulation, stress management, and interpersonal relationships in both professional and personal contexts. (2) LKM Practice (30 min): Participants engaged in guided meditation exercises incorporating focused attention, visualization of receiving and extending compassion, and practicing empathy toward individuals with whom they had experienced conflict. These exercises aimed to cultivate mindfulness, foster a compassionate mindset, and promote positive attitudes toward public service. (3) Reflection and Journaling (15 min): After each session, participants recorded their thoughts and experiences in a digital journal, reflecting on how LKM influenced their emotions and interactions. To reinforce the intervention’s impact, they were encouraged to apply LKM principles in their daily lives through informal compassion-based exercises.

#### 2.3.2. Compliance and Tracking

To encourage adherence, participants received daily reminders through the app and were prompted to log their session completion each day. The research team monitored completion rates, and participants who completed at least 80% of the sessions over the 6-week period were considered fully compliant. Additionally, participants completed a weekly reflection journal in the app, documenting their experiences, challenges, and perceived changes over the course of the intervention ([91]).

#### 2.3.3. Facilitator Support

Although the intervention was primarily app-guided, participants had access to a facilitator via a chat function within the app. The facilitator, a certified mindfulness practitioner, provided support by answering questions, offering additional resources, and encouraging participants to reflect on their progress and challenges. This support was designed to enhance engagement and deepen the impact of the LKM practice by providing personalized guidance throughout the intervention ([78]).

#### 2.3.4. Usability Evaluation

To assess the usability and acceptance of the short video app intervention, participants in the LKM group completed a post-intervention survey evaluating their views on the short video intervention. The survey used a self-report questionnaire that covered the usability, user-friendliness, emotional experience, and stress-relief effects of the intervention. The questionnaire was scored using a dichotomous scale (1 = Yes, 2 = No) and included several open-ended questions that allowed participants to provide additional feedback. The specific items and results are shown in Table 2.

### 2.4. Procedures and Design

The Loving-Kindness Meditation (LKM) intervention was delivered via a mobile application, enabling participants to engage in guided meditation sessions individually on their smartphones. These include not only guided LKM courses, but also psychoeducational materials, self-reflection exercises, and interactive engagement functions-additional components designed to enhance the learning experience and support participants’ emotional regulation, stress management, and professional interaction. Prior to the intervention, all participants attended a virtual onboarding session facilitated by the research team. This session provided an overview of the study’s objectives, ([105]) detailed instructions on app navigation and a demonstration of how to log meditation activities. Additionally, participants had the opportunity to ask questions and familiarize themselves with the app’s interface.

#### 2.4.1. The Intervention Lasted for Six Weeks and Consisted of the Following Structured Components

Daily Meditation Sessions: Participants completed 10 min guided LKM sessions daily through the mobile application ([68]).Weekly Extended Sessions: In addition to daily practice, participants engaged in one 30 min meditation session each week to deepen their practice ([29]).Reminders and Facilitator Support: To encourage adherence, the app sent automated daily reminders. Furthermore, a mindfulness facilitator was available through the app’s chat function to provide ongoing support and answer any participant inquiries ([16]).Customization for Law Enforcement Officers: The LKM content was specifically tailored to the law enforcement context, incorporating real-life scenarios relevant to police officers, such as fostering compassion during high-stress situations, enhancing resilience in the face of job-related challenges, and managing interpersonal conflicts with empathy ([21]).

#### 2.4.2. Adherence Monitoring and Engagement Metrics

To assess engagement and adherence, multiple tracking mechanisms were implemented:

Application Log Data: The app recorded log-in timestamps and session completion rates to monitor participant engagement ([95]).

Dwell Time Analysis: The duration of each session was logged to determine whether participants completed the full meditation exercises ([69]). Post-Session Journaling: Participants were required to submit a brief reflection journal entry following each session. Completion rates for these entries were analyzed as an additional measure of adherence ([44]).

Incentives for Participation: To promote sustained engagement throughout the intervention, participants who successfully completed at least 80% of the daily meditation sessions and all weekly sessions received a digital certificate of completion ([29]). Additionally, they were entered into a draw for professional development training credits, an incentive particularly relevant for law enforcement personnel ([16]).

Please refer to Table 3 for detailed information.

## 3. Results

To assess the feasibility and acceptability of the intervention, participants completed a post-intervention survey evaluating their perceptions of the LKM program. The results indicated that 83% of participants found the short-video format to be user-friendly and convenient for integration into their daily routines. Furthermore, 79% of participants reported enjoying the guided LKM sessions, citing increased emotional awareness and stress relief as primary benefits. Regarding usability, 76% of participants rated the intervention as highly practical for law enforcement professionals, and 72% stated they would recommend it to their colleagues. Open-ended responses further revealed that participants appreciated the intervention’s accessibility, though some suggested that additional interactive elements, such as real-time discussions or live coaching sessions, could enhance engagement.

In this study, a two-arm randomized controlled trial design of 2 (Group: LKM, Control) × 2 (Time: Pre, Post) was adopted, and MANOVA was used to examine the different effects of experimental intervention on MAAS, PSM, CRS, and CSS scores of subjects in the two groups. Descriptive statistics of the results are shown in Table 4.

The results of the multivariate tests showed that there was a significant main effect on the Group: F(4,177) = 13.363, *p* < 0.001, η^2^ = 0.232; a significant main effect on Time: F(4,177) = 6.440, *p* < 0.001, η^2^ = 0.127; and a significant interaction effect between Group and Time: F(4,177) = 15.572, *p* < 0.001, η^2^ = 0.260. The results of the Between-Subjects Effects tests are shown in Table 5.

The correlations of the measured variables are presented in Table 6.

From the results of Table 3, there were significant interaction effects between Group and Time in the three measures (PSM, CRS, and CSS). These interaction effects are shown in more detail in Figure 2.

It can be seen from the above results that the LKM experimental intervention significantly improves the level of subjects’ public service motivation, conflict resolution skills, and communication skills. Therefore, all the hypotheses were supported. However, it is worth noting that LKM intervention did not significantly improve the level of mindfulness of participants in the LKM group.

The effect size for CSS (η^2^ = 0.214) is particularly noteworthy, exceeding the average effect sizes found in previous research on law enforcement personnel ([67]). This may suggest that LKM’s emphasis on fostering interpersonal compassion enhances communication effectiveness beyond general mindfulness training. However, the effect sizes for PSM (η^2^ = 0.108) and CRS (η^2^ = 0.104) fall within the expected range based on prior research ([21]). These findings highlight the potential of LKM for improving communication skills in high-stress professions but also indicate that future interventions might benefit from incorporating complementary training, such as breath awareness meditation, to further enhance mindfulness-related outcomes ([51]).

### Adherence and Its Association with Outcomes

Among participants in the LKM group, adherence was high: 87% completed at least 80% of the daily sessions, and 91% completed all six weekly 30 min sessions. To further understand the role of adherence in intervention outcomes, Pearson correlation analyses were conducted between adherence rate and the change scores (post-pre) of PSM, CRS, and CSS.

The results revealed significant positive correlations between adherence rate and improvements in:

Public service motivation (r = 0.42, *p* < 0.01), conflict resolution skills (r = 0.38, *p* < 0.01), and communication skills (r = 0.45, *p* < 0.001).

These findings suggest that higher engagement with the LKM intervention was associated with greater improvements in professional competencies.

## 4. Discussion

### 4.1. LKM’s Lack of Impact on Mindfulness

Despite the similarities between mindfulness and Loving-Kindness Meditation (LKM), such as fostering emotional regulation and self-awareness, these two practices have distinct focal points and mechanisms. According to Self-Determination Theory (SDT), the development of mindfulness often requires a high level of intrinsic motivation and a commitment to autonomous, sustained practice ([19]). Mindfulness meditation specifically involves cultivating present-moment awareness, a process that demands ongoing, intentional engagement with one’s thoughts and sensations. In contrast, LKM emphasizes the cultivation of compassion and positive regard for others, focusing more on building empathy than on present-moment awareness ([64]). While mindfulness involves moment-to-moment awareness without judgment, LKM primarily enhances emotional connectivity, which may not directly translate into improved mindfulness in the traditional sense. For police officers who may not have prior training in mindfulness techniques, LKM alone may not be sufficient for cultivating mindfulness, as they may lack the specific focus and techniques that mindfulness training requires.

From the perspective of Social Learning Theory (SLT), the development of mindfulness requires establishing habitual behaviors through continuous practice rather than solely cultivating compassion ([66]). Mindfulness is a skill that typically requires regular reinforcement and specific mental exercises focused on observing and accepting one’s thoughts and emotions without judgment ([67]). LKM, delivered in short video format, may lack the depth and structure necessary to foster these mindfulness skills effectively ([23]). Traditional mindfulness training often involves guided, structured sessions that help practitioners build consistent habits of present-moment awareness, a skill set that requires professional guidance and sustained practice to fully develop ([38]). Given that the LKM sessions in this study were relatively brief and delivered via short video formats, the intervention may not have provided sufficient engagement to facilitate substantial changes in mindfulness. Future research could explore hybrid interventions that combine LKM with structured mindfulness training to examine whether this integration enhances both compassion and mindfulness. As a result, while LKM can enhance emotional regulation through compassion, it may not provide the necessary tools for cultivating the specific qualities of mindfulness.

Furthermore, compared with other studies, the LKM intervention in this study was delivered through short video-guided sessions rather than in-person instruction or extended meditation practice ([80]). Some studies have indicated that short-term LKM training may be more effective in improving emotional regulation and empathy, but may require additional training support for fostering long-term mindfulness awareness. Future research could consider integrating LKM with traditional mindfulness meditation (such as breath awareness training) or extending the intervention duration to assess whether it can significantly enhance mindfulness.

In conclusion, this study found that LKM effectively improved police officers’ public service motivation, conflict resolution skills, and communication abilities, but its impact on mindfulness was relatively limited. Future studies could further explore the effects of combining LKM with other mindfulness training approaches or examine how individual baseline characteristics influence the effectiveness of LKM in enhancing mindfulness ([18]).

### 4.2. LKM’s Effect on Public Service Motivation (PSM)

The current study found that Loving-Kindness Meditation (LKM) significantly enhanced public service motivation (PSM) among police officers, a finding that can be explained through Self-Determination Theory (SDT) and Social Learning Theory (SLT) ([65]). According to SDT, PSM is driven by both intrinsic and extrinsic motivation, where intrinsic motivation is strengthened through personal values and the fulfillment of psychological needs. LKM, with its emphasis on compassion and empathy, nurtures these intrinsic motivations by fostering a sense of connectedness and empathy toward others ([64]). This sense of connectedness aligns with SDT’s emphasis on relatedness, one of the three basic psychological needs essential for maintaining intrinsic motivation. Moreover, prior research suggests that self-compassion plays a crucial role in maintaining motivation in high-stress professions. By fostering self-kindness rather than self-criticism, LKM may protect police officers from emotional exhaustion, allowing them to sustain their motivation to serve the public ([78]). For police officers, a profession characterized by public service and community interaction, LKM cultivates compassion toward others, thereby reinforcing their motivation to serve the public and fulfill their role with greater dedication ([19]).

Recent research supports the role of compassion in enhancing intrinsic motivation for public service. A study by [71] ([71]) demonstrated that prosocial behaviors and compassionate mindsets increased intrinsic motivation by fulfilling the need for relatedness, which, according to SDT, is crucial for well-being and self-motivation. Similarly, [97] ([97]) found that when individuals engage in activities that align with their values and promote compassion, their motivation to contribute to the well-being of others increases. For police officers practicing LKM, these compassionate values become internalized, enhancing their sense of responsibility and altruism, which are essential components of PSM.

From an SLT perspective, LKM guided through short video apps provides officers with a model of prosocial and compassionate behaviors, which they can observe and emulate. According to SLT, people learn behaviors through observation, imitation, and modeling, especially when they perceive the observed behaviors as beneficial or aligned with their values ([8]). By incorporating real-world policing scenarios into LKM training videos, future interventions could enhance the relevance of observed behaviors and increase their applicability to law enforcement settings.

In this case, video-guided LKM offers officers the opportunity to witness acts of empathy and kindness, thus reinforcing prosocial behaviors that can translate to their professional roles. Studies have shown that video-based interventions are effective in behavior modeling and attitude change, particularly when the content is prosocial and promotes values aligned with the viewer’s responsibilities ([38]).

Moreover, LKM allows police officers to cognitively reframe their roles, fostering a deeper understanding of their impact on society. By engaging in LKM, officers develop a sense of empathy and shared humanity, which can positively reshape their perception of their profession. This aligns with findings suggesting that self-compassion can buffer against compassion fatigue and emotional burnout in high-stress professions, allowing individuals to sustain prosocial motivation over time ([41]).

This cognitive reframing aligns with SLT’s concept of self-efficacy and internalized motivation, where individuals begin to view their roles through the lens of compassion and duty ([80]). Research by Roeser and Bonilla suggests that compassion-based practices in professional settings can increase job satisfaction and meaningful engagement by encouraging individuals to perceive their work as contributing to the greater good ([36]).

In summary, the increase in PSM among police officers practicing LKM can be attributed to the dual influences of SDT and SLT. LKM satisfies intrinsic motivational needs by fostering compassion, fulfilling the need for relatedness, and reinforcing prosocial values ([60]). Furthermore, this study highlights the role of self-compassion in supporting motivation, particularly in mitigating the emotional strain associated with public service work. Future studies could explore whether the effects of LKM on PSM are moderated by levels of self-compassion, providing a more comprehensive understanding of how these mechanisms interact.

Additionally, by providing a model of compassionate behavior, LKM empowers officers to internalize these values, motivating them to serve the public with a renewed sense of purpose and commitment ([30]).

### 4.3. LKM’s Positive Influence on Conflict Resolution Skills

The study found that Loving-Kindness Meditation (LKM) significantly improved police officers’ conflict resolution skills. This outcome can be explained through Self-Determination Theory (SDT) and Social Learning Theory (SLT) ([20]). According to SDT, emotional regulation and interpersonal communication are essential elements of self-directed, autonomous behavior, and they play a crucial role in conflict resolution. LKM promotes empathy and emotional regulation, equipping police officers with the psychological tools to manage conflict situations more effectively ([66]). Through the cultivation of compassion, officers are better able to maintain calmness and emotional stability, reducing the likelihood of impulsive or aggressive responses during conflict. This aligns with the SDT’s view that fulfilling the psychological needs of relatedness and emotional regulation promotes constructive interactions and enhances interpersonal skills ([64]).

LKM facilitates a mindset that enables officers to empathize with conflicting parties, which is essential for conflict resolution. The practice of generating compassion and kindness helps officers approach conflicts with a more open and understanding attitude, enhancing their capacity for perspective-taking. Empathy, as cultivated through LKM, provides a foundation for improved interpersonal relationships and can reduce defensive behaviors, which are often triggers for escalating conflicts ([65]). Additionally, by focusing on non-judgmental awareness, LKM practitioners are more likely to assess situations calmly, allowing them to find constructive solutions in high-stress interactions, such as those encountered in police work.

From the perspective of Social Learning Theory (SLT), LKM video guidance provides police officers with constructive conflict resolution models, which are critical for observational learning. SLT posits that individuals learn behaviors by observing and imitating role models, particularly when these models demonstrate behaviors that result in positive outcomes. Through video-guided LKM, police officers can observe examples of conflict resolution strategies, such as active listening, empathy, and seeking mutual understanding. By repeatedly watching these behaviors, officers are likely to internalize and replicate these constructive skills in their own interactions, leading to more effective and peaceful conflict resolution.

Empirical evidence supports the efficacy of compassion-based interventions like LKM in enhancing conflict resolution skills. For instance, a study by Eckland, Leyro, Mendes and Thompson ([31]) found that compassion meditation improved emotional regulation and empathy, both of which contribute to conflict management. Furthermore, [63] ([63]) demonstrated that LKM practitioners were better able to navigate interpersonal conflicts due to an increased capacity for empathy and non-reactivity. These findings suggest that integrating LKM into police training can be an effective approach to equipping officers with essential conflict resolution skills. By fostering both emotional regulation (through SDT mechanisms) and observational learning (through SLT), LKM enables police officers to approach conflicts with empathy, reducing the likelihood of escalation and supporting community trust ([66]).

### 4.4. LKM’s Positive Effect on Communication Skills

The study revealed that Loving-Kindness Meditation (LKM) significantly improved communication skills among police officers. This effect can be understood through Self-Determination Theory (SDT) and Social Learning Theory (SLT). According to SDT, effective communication is driven by intrinsic motivation, where individuals are motivated to connect meaningfully with others. LKM nurtures this intrinsic motivation by fostering empathy and a non-judgmental attitude, which encourages open, patient, and genuine communication ([64]). By cultivating compassion toward oneself and others, LKM allows officers to engage with the public in a manner that prioritizes understanding and connection, essential aspects of effective communication in high-stress environments such as policing ([36]).

Empathy is central to SDT as it satisfies the need for relatedness, one of the theory’s core psychological needs, which plays a key role in fostering prosocial behavior and improving interpersonal skills ([88]). Through LKM, officers learn to approach others with kindness and empathy, which can lead to more attentive listening and thoughtful responses. This development of empathy and non-judgmental awareness aligns with SDT’s premise that fulfilling psychological needs enhances self-motivation and supports prosocial behavior, which is critical for effective communication ([101]). Thus, LKM serves as a tool to strengthen police officers’ intrinsic motivation to communicate constructively and patiently with community members.

Social Learning Theory also provides a valuable perspective on how LKM enhances communication skills. SLT posits that individuals acquire new behaviors through observation and imitation of others, especially when these behaviors lead to positive outcomes ([2]). LKM, when delivered through video guidance, exposes officers to positive communication models, such as empathic listening, calm expression, and perspective-taking. By repeatedly observing these examples, police officers can internalize and adopt these behaviors in their daily interactions with the public. The video-based format of LKM allows officers to see concrete examples of constructive communication, reinforcing strategies that can prevent misunderstandings, foster trust, and promote cooperation ([18]).

Recent studies support the effectiveness of compassion and mindfulness-based interventions, like LKM, in enhancing communication skills. For instance, a study by Malin found that compassion training improved participants’ interpersonal communication by increasing emotional awareness and reducing defensive responses ([20]). Similarly, a study by Eckland, Leyro, Mendes and Thompson demonstrated that empathy-based training enhanced individuals’ ability to communicate openly and attentively, which are critical skills in professions involving frequent public interaction. By combining the empathy-focused approach of LKM with observational learning through SLT, police officers can cultivate the skills necessary for building trustful relationships and effectively navigating complex social interactions ([66]).

In summary, LKM’s emphasis on empathy, non-judgmental awareness, and positive role modeling aligns with the principles of SDT and SLT, facilitating improvements in communication skills ([101]). Through intrinsic motivation (SDT) and observational learning (SLT), LKM provides police officers with the psychological tools and behavioral models needed to communicate effectively, fostering more constructive and respectful interactions with the public. This enhancement in communication skills may contribute to a more positive public perception of police officers, reinforcing community trust and cooperation ([65]).

### 4.5. Limitations and Future Research

This study was conducted within a single metropolitan police department in China, which may limit the generalizability of the findings to other law enforcement agencies or cultural contexts ([103]). However, single-site studies are commonly used in police training research to maintain consistency in intervention delivery and organizational culture ([47]). Expanding the study to multiple police departments or including officers from different regions could enhance the external validity of the findings ([21]). Future research could employ a stratified sampling approach to capture potential variations in organizational norms, stress exposure, and departmental policies affecting mindfulness-based interventions ([65]). Additionally, cross-cultural studies could further examine whether the effects of LKM are influenced by different societal attitudes toward meditation and emotional regulation ([21]).

Second, the study’s sample was limited to police officers from a single metropolitan police department in China, which may restrict the generalizability of the findings to other law enforcement contexts or cultural settings ([87]). Differences in organizational culture, societal expectations, and baseline levels of mindfulness could influence how LKM interventions impact police officers in other regions. Future research should examine LKM interventions across diverse geographic locations and law enforcement agencies to determine whether these effects are consistent in varied settings ([98]).

Third, the intervention relied on a short video format to deliver LKM practices, which, while convenient, may lack the depth and intensity of traditional in-person or longer guided meditation sessions. This format may not be as effective in cultivating sustained changes in mindfulness or empathy, particularly for individuals with limited prior experience in meditation practices ([36]; [51]). Future research should explore variations in intervention delivery, such as integrating in-person sessions or providing longer, more immersive LKM practices, to assess whether these modifications produce stronger or more lasting effects ([80]).

Fourth, while this study utilized validated self-report scales to assess mindfulness, public service motivation, conflict resolution skills, and communication skills, we acknowledge the potential limitations associated with response biases ([84]). Self-reported measures, while widely used, rely on participants’ perceptions and may not fully capture real-world behavioral changes. Future studies should incorporate behavioral observations, peer assessments, and video/audio analyses to provide a more comprehensive evaluation of intervention effects ([56]). For instance, conflict resolution skills could be assessed through role-play simulations with trained evaluators rating officers’ responses to de-escalation scenarios ([45]). Similarly, video or audio recordings of police interactions could be analyzed using validated communication assessment frameworks ([33]). Additionally, peer evaluations could serve as an alternative method to cross-validate findings ([101]). By integrating these objective measures, future research could strengthen the reliability of findings and provide deeper insights into the practical applications of LKM interventions in law enforcement settings.

Fifth, additionally, this study did not directly measure perceived stress and job satisfaction, both of which are critical indicators of occupational well-being in high-stress professions such as law enforcement ([57]). Prior research suggests that mindfulness-based interventions, including LKM, may contribute to stress reduction and increased job satisfaction by promoting emotional regulation and resilience ([61]). Future studies should explicitly assess these variables to determine whether LKM has indirect benefits beyond interpersonal skills, particularly in mitigating occupational stress and enhancing officers’ overall job fulfillment ([99]). Longitudinal research could further explore whether reductions in stress contribute to sustained improvements in motivation, conflict resolution, and communication abilities over time ([107]).

Sixth, the findings indicate a high level of acceptance and perceived usability of the LKM intervention among police officers, supporting its feasibility as a digital mindfulness intervention ([64]). The willingness of participants to recommend the program suggests its practical relevance to law enforcement personnel ([4]). However, future studies could further explore interactive features to enhance engagement, such as real-time facilitator support or peer discussion forums ([18]). Additionally, longitudinal follow-up is necessary to determine whether the intervention’s benefits persist over time ([17]).

Seventh, while this study provides valuable insights into the effects of Loving-Kindness Meditation (LKM) on police officers, several limitations should be acknowledged. First, the study did not examine potential mediators (e.g., emotional regulation, empathy) or moderators (e.g., personality traits, baseline stress levels), which could help explain individual differences in intervention outcomes ([49]). Future research should explore these factors to clarify the mechanisms driving LKM’s effects ([50]). Second, although the study used a randomized controlled design, additional control variables (e.g., workload, job tenure, baseline emotional resilience) could further improve causal inference ([84]). Finally, this study assessed outcomes only immediately after the intervention, limiting conclusions about the long-term sustainability of the observed effects ([1]). Future studies should incorporate extended follow-up assessments (e.g., six months or one year post-intervention) to determine whether LKM’s benefits persist over time ([52]). Addressing these limitations will enhance the robustness and applicability of LKM interventions for law enforcement professionals.

Eighth, another important consideration is the potential influence of peer discussions among participants. Given the nature of police work, officers may have shared their experiences with colleagues, which could have influenced perceptions of the intervention. While this study did not explicitly track such discussions, future research could incorporate qualitative methods, such as interviews or focus groups, to explore how peer interactions shape engagement and outcomes. Additionally, self-selection bias should be acknowledged, as officers voluntarily opted into the study. This may have resulted in a sample skewed toward individuals already inclined toward mindfulness or personal development. Future studies could address this by employing mandatory training formats or stratified sampling to ensure a more representative participant pool. These considerations will provide a more nuanced understanding of how LKM interventions function in real-world law enforcement settings.

Another limitation of this study is the relatively short duration of the intervention (6 weeks) and the lack of long-term follow-up to examine the persistence of effects. While the results suggest immediate post-intervention benefits, it is unclear whether these improvements in mindfulness, public service motivation, conflict resolution, and communication skills are sustained over time ([51]). Future studies could incorporate follow-up assessments at six months or one year post-intervention to evaluate the longevity of LKM’s impact on police officers ([56]).

Ninth, this study employed a wait-list control rather than an active or placebo control task. Although this design isolates the effects of the LKM-based videos, it does not fully rule out expectancy or Hawthorne effects. Future research should incorporate an active comparison condition—e.g., neutral skills-training videos of equal duration—to more rigorously control for nonspecific intervention effects.

Tenth, while this study provides initial evidence for the effectiveness of short video app-guided LKM in improving psychological and professional outcomes among police officers, future research should move beyond outcome verification to examine how and for whom such interventions work. In particular, we recommend investigating self-compassion as a potential mediating mechanism through which LKM exerts its effects, as well as exploring whether factors such as baseline occupational stress, trait empathy, or digital receptivity moderate the intervention’s impact. These directions would help to clarify both the process and boundary conditions of LKM’s influence in high-stress occupational settings.

Lastly, this study did not examine potential mediators or moderators that might influence the effectiveness of the LKM intervention ([5]). Factors such as baseline levels of empathy, stress, or prior exposure to mindfulness practices could affect individual responsiveness to the intervention. Additionally, personality traits or levels of intrinsic motivation may moderate the degree of improvement in the targeted skills ([39]). Future research should investigate these potential mediators and moderators to better understand which individuals benefit most from LKM and under what conditions it is most effective ([100]). Future research should incorporate a systematic investigation of these potential mediators and moderators to refine the theoretical framework underpinning LKM’s effects. Employing structural equation modeling (SEM) or longitudinal designs could help disentangle these complex relationships and provide a more nuanced understanding of how LKM operates within different individual and organizational contexts ([48]). This would be particularly valuable in tailoring LKM interventions to diverse populations, ensuring their effectiveness across varying professional and personal backgrounds ([36]).

In conclusion, while this study highlights the potential of short video app-guided LKM to improve police officers’ interpersonal and professional skills, future research should address these limitations by incorporating objective assessments, expanding the sample to different settings, exploring alternative delivery methods, conducting long-term follow-ups, and examining individual differences in intervention responsiveness. Addressing these aspects will contribute to a more comprehensive understanding of LKM’s applicability and effectiveness in law enforcement and other high-stress professions ([27]).

## 5. Conclusions

This study provides empirical support for using short video app-guided Loving-Kindness Meditation (LKM) to improve mindfulness, self-compassion, and professional competencies among police officers. These findings offer practical value for law enforcement agencies seeking scalable, low-cost mental health interventions. Integrating LKM into routine training may help reduce stress, enhance communication, and foster a stronger service orientation. The results underscore the real-world potential of compassion-based practices to promote both officer well-being and public service effectiveness.

## Figures and Tables

**Figure 1 behavsci-15-00909-f001:**
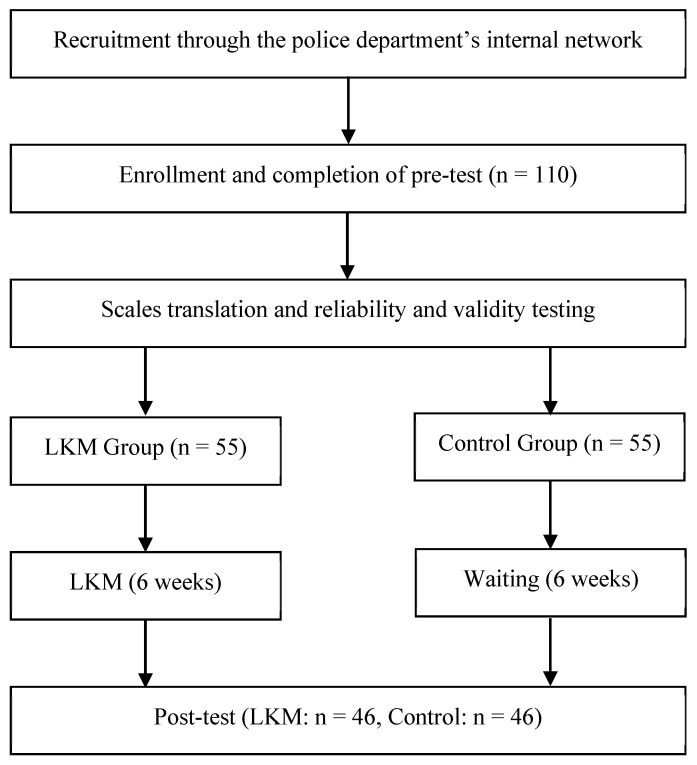
Procedure flow chart.

**Figure 2 behavsci-15-00909-f002:**
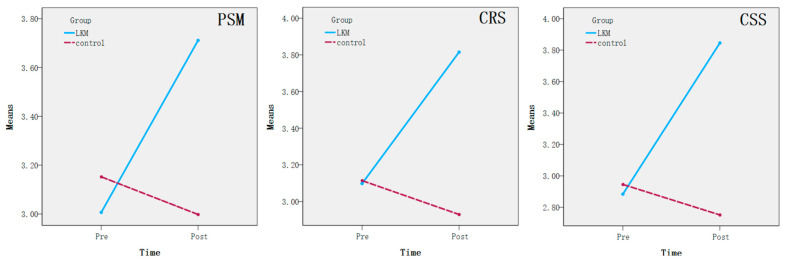
Significant interaction effects between Group and Time. Note. PSM: Public service motivation; CRS: conflict resolution skills; CSS: Communication Skills Scale.

**Table 1 behavsci-15-00909-t001:** Baseline demographic characteristics of participants.

Characteristic	Total (n = 92)	LKM Group (n = 46)	Control Group (n = 46)
Age (Mean ± SD)	35.22 ± 3.19	35.57 ± 3.30	34.86 ± 3.02
Gender			
Male (%)	66 (71.7%)	32 (69.6%)	34 (73.9%)
Female (%)	26 (28.3%)	14 (30.4%)	12 (26.1%)
Years of Police Service			
1–5 years (%)	28 (30.4%)	15 (32.6%)	13 (28.3%)
6–10 years (%)	41 (44.6%)	21 (45.7%)	20 (43.5%)
Over 10 years (%)	23 (25.0%)	10 (21.7%)	13 (28.3%)
Prior Mindfulness Experience (%)	23 (25.0%)	11 (23.9%)	12 (26.1%)

**Table 2 behavsci-15-00909-t002:** Usability Evaluation.

Items	Yes n (%)	No n (%)
1. User-Friendliness and Convenience: Do you think the short video app is easy to use and convenient to integrate into daily life?	38 (83%)	8 (17%)
2. Emotional Awareness and Stress Relief: Do you think the short video app-guided LKM exercise is helpful in improving emotional awareness and reducing stress?	36 (78%)	10 (22%)
3. Usability and Practicality: Do you think the short video app-guided LKM exercise meets the work needs of law enforcement personnel?	35 (76%)	11 (24%)
4. Promotability: Would you recommend this short video app-guided LKM exercise to your colleagues?	33 (72%)	13 (28%)
Open-ended Item	Response Summary	
What other suggestions do you have for short video app-guided LKM exercises?	Add more interactive elements, include discussions or online coaching to increase engagement	

**Table 3 behavsci-15-00909-t003:** Scope and sequence of the six-week intervention.

Week	Session Content & Focus
1	Introduction to Loving-Kindness Meditation (LKM); psychoeducation on its psychological benefits, focusing on emotional regulation and interpersonal relationships.
2	Practicing LKM with attention to empathy and compassion towards self and others; mindfulness in stressful situations.
3	LKM for enhancing resilience; extending compassion in conflict situations, particularly in professional settings.
4	Exploring emotional regulation through LKM; managing stress and anxiety in law enforcement contexts.
5	Deepening LKM practice with focus on interpersonal communication; fostering positive attitudes toward colleagues and the community.
6	Integrating LKM into daily life; reflecting on the application of compassion-based exercises to enhance overall well-being and professional effectiveness.

**Table 4 behavsci-15-00909-t004:** Descriptive statistics.

Group	Measures	Mean (SD)	
		Pre	Post
LKM (n = 46)	MAAS	3.574(0.525)	3.872(0.666)
	PSM	3.007(0.565)	3.711(0.690)
	CRS	3.098(0.581)	3.815(0.690)
	CSS	2.884(0.496)	3.845(0.629)
Control (n = 46)	MAAS	3.630(0.596)	3.609(0.761)
	PSM	3.152(0.485)	2.998(0.725)
	CRS	3.114(0.555)	2.929(0.818)
	CSS	2.946(0.460)	2.752(0.630)

Note. MAAS: Mindfulness Attention Awareness Scale; PSM: public service motivation; CRS: conflict resolution skills; CSS: Communication Skills Scale.

**Table 5 behavsci-15-00909-t005:** Between-Subjects Effects tests.

Measures	Variable	F	*p*	η^2^
MAAS	Group	1.193	0.276	0.007
	Time	2.128	0.146	0.012
	Group × Time	2.850	0.930	0.016
PSM	Group **	9.515	0.002	0.050
	Time **	8.941	0.003	0.047
	Group × Time ***	21.793	<0.001	0.108
CRS	Group ***	19.435	<0.001	0.097
	Time **	7.291	0.008	0.039
	Group × Time **	20.920	<0.001	0.104
CSS	Group ***	39.182	<0.001	0.179
	Time ***	21.672	<0.001	0.107
	Group × Time ***	49.095	<0.001	0.214

Note. MAAS: Mindfulness Attention Awareness Scale; PSM: public service motivation; CRS: conflict resolution skills; CSS: Communication Skills Scale; ** *p* < 0.01; *** *p* < 0.001. The confidence interval is 95%.

**Table 6 behavsci-15-00909-t006:** Correlation between the study variables.

	1	2	3	4
MAAS	—			
PSM	0.678 ***	—		
CRS	0.628 ***	0.740 ***	—	
CSS	0.604 ***	0.771 ***	0.806 ***	—

Note. MAAS: Mindfulness Attention Awareness Scale; PSM: public service motivation; CRS: conflict resolution skills; CSS: Communication Skills Scale; *** *p* < 0.001.

## Data Availability

The original contributions presented in the study are included in the article. Further inquiries can be directed to the corresponding author. The data presented in this study are available on request from the first author due to privacy restrictions.

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
