# Peer review of "The Effects of Loving-Kindness Meditation Guided by Short Video Apps on Policemen’s Mindfulness, Public Service Motivation, Conflict Resolution Skills, and Communication Skills"

_behavsci, 2025, doi:10.3390/bs15070909_

Round 1
Reviewer 1 Report
Comments and Suggestions for Authors
Literature review
It would be helpful for the authors to describe the key concepts of self-compassion and mindfulness in more detail as they are key to the study. Also include a discussion on there relationship.
Methods
Please elaborate more on the measures. Do they sub scales? Give some examples of individual items on the scales, etc
There are several measures of mindfulness. Perhaps the authors could elaborate on why they used the MAAS compared to other ones (like the five facets of mindfulness). This might also to mention this in the limitations as this study focused on only one aspect of mindfulness.
Include more details about the control group. Were there any alternate tasks or placebos? If not add a discussion to the limitations section
Reviewer 2 Report
Comments and Suggestions for Authors
This manuscript presents an engaging and relevant study. The design is well-conceived and the writing generally clear. However, several conceptual and structural aspects require attention before publication.
I encourage the authors to critically revisit the theoretical framing of their claims in the introduction, ensuring all major assertions are supported by relevant citations. In its current form, the literature review includes several broad or declarative statements that would benefit from clearer sourcing and justification. Similarly, the hypotheses would be stronger if framed with greater theoretical nuance, particularly by introducing potential mediating or moderating mechanisms.
Finally, I would recommend strengthening the conclusion by moving beyond general restatements or future research directions. Instead, focus on the practical implications of the findings, which have clear relevance to training, public service interventions, and digital well-being initiatives in high-stress professions.
Refined In-Text Comments
- Abstract: The abstract is generally strong. However, I would suggest concluding with the real-world implications of the findings rather than recommendations for future research, which are more appropriate for the discussion section.
- Line 47–48: While the claim may be broadly acceptable, assertions of this kind should be supported with references. Scientific writing benefits from grounding even seemingly intuitive statements in existing literature.
- Line 48–50: The same applies here. Please support claims with appropriate citations.
- Line 55: The reliance on a single study from 2019 weakens the point being made. A broader engagement with both older and more recent literature would strengthen the claim considerably.
- Lines 56–57: This statement lacks attribution. Please clarify who has made this claim or on what basis it is being presented.
- Lines 58–59: As above, this claim would benefit from citations. As it stands, it appears speculative.
- Lines 59–62: A single reference is insufficient to substantiate the broader argument being made. I recommend engaging in more critical dialogue with other relevant literature.
- Line 62–63: The 2013 source may still be relevant, but the claim would be more persuasive if supported by more recent and complementary studies.
- Lines 67–70: Please provide supporting references for these assertions.
- Lines 72–73: The claim requires citation or further explanation.
- Line 76: A reference is needed here to support the claim.
- Lines 76–77: This general statement would benefit from some empirical backing. Furthermore, elaborating on why these professions are considered particularly stressful could provide important context, particularly if these stressors are central to the rationale of the study.
- Line 86–88: If this constitutes a central argument of the paper, it would be helpful to clarify and strengthen it here.
- Lines 101–103: The identification of a research gap feels underdeveloped. The argument requires further elaboration to convincingly establish the novelty or necessity of the study.
- Line 109: This is not the most appropriate place to outline future research directions. If the authors wish to retain this point, I suggest creating a dedicated section in the conclusion.
- Lines 111–113: This statement may appear overly self-evident. Consider reframing or removing it to avoid redundancy.
- Line 120: There appears to be a formatting issue (extra spacing).
- Line 124: The summary of Loving-Kindness Meditation (LKM) would benefit from more detailed sourcing. It’s not entirely clear which aspects are grounded in the literature.
- Lines 181–188 (Hypotheses):
The hypotheses appear overly straightforward, largely reiterating the aims of the study without engaging with potential mechanisms or nuances. For instance, stating that LKM will improve outcomes such as mindfulness or communication skills is predictable given the design. I would encourage the authors to consider more theoretically driven hypotheses, such as whether self-compassion mediates these effects or whether certain baseline characteristics moderate the intervention’s impact. This would elevate the contribution from a confirmation of efficacy to a deeper understanding of how and for whom such interventions work. - Conclusion: The conclusion comes across as rather general. While the findings are summarised, more attention could be given to their applied implications. There is some mention of practical significance, but it could be expanded to better inform policy, training programmes, or organisational interventions. Avoid ending with vague calls for more research—this is the moment to articulate the significance of the current findings.
